# Measuring respectful maternal and newborn care in Nepal: Comparing linked observation and interview data- prospective cohort study

Omkar Basnet[1], Emma Sacks[2], Mary V. Kinney[3,4], Ashish KC[5,6]*

1 Research Division, Golden Community, Lalitpur, Nepal, 2 Department of International Health, John Hopkins School of Public Health, Baltimore, United States of America, 3 School of Public Health, University of the Western Cape, Belville, South Africa, 4 Division of Global Surgery, University of Cape Town, Cape Town, South Africa, 5 School of Public Health and Community Medicine, Institute of Medicine, Sahlgrenska Academy, University of Gothenburg, Gothenburg, Sweden, 6 Department of Women's and Children's Health, Faculty of Medicine, Uppsala University, Uppsala, Sweden

* ashish.kc@gu.se

## Abstract

### Background

Respectful maternal and newborn care is the cornerstone of high-quality care, however, measuring experience of respectful care has challenges since it can be subjective, and dependent on expectations. In this study, we assess the concordance between women's reported experiences of respectful maternal and newborn care and independent observation of their care in Nepal.

### Methods

This is a secondary analysis of a prospective cohort study among 22832 pregnant women conducted in three high volume hospitals in the country: Koshi Provincial Hospital (Hospital A), Bharatpur Hospital (Hospital B), and Lumbini Provincial Hospital (Hospital C) for 18 months between April 2017 and October 2018. The study implemented direct observation during and semi-structured interviews at discharge to evaluate the quality of maternal and newborn care in three large public hospitals. For this analysis, three domains for respectful maternal and newborn care were considered: 1) consent and counselling 2) respect and dignity of care, and 3) care provision. The two data sources (observation checklist and semi-structured interview) were plotted to these three domains to identify common indicators. The level of agreement (LOA) between two measurements was compared using Cohen kappa scores ($\kappa$) and Bland Altman plots.

### Findings

During the study period, 22832 women had both observation and interview completed. For consent and counseling, 78.8% of women reported being informed about

**Data availability statement:** All relevant data are within the paper and its Supporting information files.

**Funding:** The authors received no specific funding for this work.

**Competing interests:** The authors have declared that no competing interests exist.

routine care while only 47.3% were observed to have been consented and counseled (k, LOA = 59.1%). For respect and dignity of care, 99.0% of women reported being treated with dignity and respect and 96.4% were observed (k, LOA = 95.4%). For care provision, 37.9% reported that the infant was kept in immediate skin-to-skin contact after delivery while only 3.9% were observed (k, LOA = 61.7%).

## Conclusion

A significant difference existed between observed and self-reported measures of maternal and newborn care. This study highlights the need for a measurement approach that incorporates independent observations alongside self-reported data. There is also a need to further explore concordance between different sources for progress monitoring.

## Background

High-quality care in health facilities can help to prevent maternal and newborn deaths and stillbirths, and improve other health outcomes. With the majority of births now taking place in health facilities (83% worldwide), poor quality of facility-based maternal and newborn care has resulted in millions of unnecessary deaths and disabilities [1,2]. In 2015, World Health Organization (WHO) provided a framework for the quality of maternal and newborn care that expands the notion of quality beyond the provision of care to include experience of care [3]. This shift underscores the importance of ensuring not only the care provided but also that it is delivered in a way that respects and supports women and their newborns.

Respectful maternal and newborn care (RMNC) is a critical component of quality as it improves trust in health institutions [4] ensuring that women and families have a positive experience during and after childbirth [5,6], including respectful care for newborns [7,8]. However, many women experience mistreatment during labor, including inadequate information, lack of consent, obstructing the presence of a birth companion, and withholding the provision of care without the consent of the woman [8,9]. These negative experiences can discourage women from seeking health care facility for childbirth, ultimately affecting maternal and newborn health outcomes. Such incidents can lower the utilization of healthcare facilities for childbirth. Additionally, when patient satisfaction is assessed, low scores are often associated with poor provision of quality care which further reinforce the need for accurate experience measurement [10].

Providing a respectful environment for mothers and newborns may be particularly challenging in low- and middle-income countries due to resource constraints including shortage of qualified health professionals, limited equipment, outdated facility and gaps in essential services [11,12]. While there are established methods for monitoring the provision of care, measuring the experience of care empirically has many challenges [13], as experiences can be subjective and dependent on expectations [14,15].

Previous studies have shown discrepancies between reported and observed experiences of care. This includes some types of disrespect and abuse are observed, but

not reported, while other types are reported but not observed [16–18]. Despite increasing global recognition on the need to measure respectful care, research has primarily focused on maternal experiences with less attention given to newborns [19,20]. Studies focusing on newborn care have revealed incidence of poor care, including newborns left unattended, unnecessarily separated from their mothers after birth, transferred to other facilities without consent of the parents, [21,22], and unsafe early discharge of mothers and their babies due to overcrowding [23]. Research further shows that many newborns receive below the standard care, both in terms of provision of care and experiences of care [21,24]. Since maternal and newborn experiences are interlinked and can affect each other [25], there is a need to better understand how to measure the experience of maternal and newborn care.

Interviews with women and direct observation are two methods for studying experience of care; however, there is limited evidence on how these methods compare in capturing maternal and newborn experiences simultaneously. While some studies have compared interview and observation methods in maternal care [26] no research to date has examined how these approaches aligns or differs when measuring both maternal and newborn experiences together.

In Nepal, there is large variation in the reported experiences of mothers during childbirth, and the quality of care provided by healthcare workers [4,7,27]. To address these concerns, in 2017–2018, the Nepal perinatal quality improvement project (NePeriQIP) was implemented to improve the provision and experience of care during and immediately after childbirth [28]. As a part of this initiative, a prospective data collection system was designed to collect data on care during birth using pre-hospital exit interview and observation.

Using data from NePeriQIP, this study critically compares information collected through direct observation of women during labor, delivery and the immediate postpartum period with self-reported experiences from interviews. This comparison is necessary to determine whether different methods yield similar or different insights as well as to assess their implications for measuring and improving respectful maternal and newborn care.

## Methodology

*Study design*- NePeriQIP is a prospective cohort study among the women admitted in the three large referral hospitals of Nepal for childbirth. This is a secondary analysis of the perinatal dataset collected during NePeriQIP study in three public referral hospitals in Nepal for 18 months between April 2017 and October 2018. In the primary study, the data on the quality of perinatal care was collected through direct observation of the events during labor and childbirth and immediately after, as well as semi-structured interviews with women at discharge.

*Study settings*- The study was conducted in three high volume hospitals in the country: Koshi Provincial Hospital (Hospital A), Bharatpur Hospital (Hospital B), and Lumbini Provincial Hospital (Hospital C). (Table 1)

*Participant inclusion criteria*- Any pregnant woman with gestational age 22 weeks or more who was admitted to the hospital for delivery was eligible for enrollment in the primary study. Women whose pregnancy had no fetal heart sound at admission or women who were referred to operation theatre for cesarean section or assisted delivery were excluded from the study. Women in the admission room were assessed and screened for eligibility and those who were eligible were approached to be enrolled in the primary study through informed written consent. For women aged less than 18 years accent was given and their legal guardian was approached for consent. For this study purpose, women who were enrolled

**Table 1. Study setting in three hospitals.**

|  | Number of annual deliveries | C-section rate | Number of labour and delivery beds | Number of nurse midwives |
|---|---|---|---|---|
| Hospital A | 9870 | 20% | 8 | 8 |
| Hospital B | 14,560 | 30% | 12 | 12 |
| Hospital C | 12,323 | 25% | 12 | 11 |

in the primary study who had both an observation during childbirth and semi-structured interview were considered for analysis.

*Data collection-* A team of two trained data collectors were stationed in the delivery room and postnatal ward as a labor and postnatal surveillance team. The labor surveillance team collected data on the labor and childbirth process using an observation checklist, while the postnatal surveillance team collected data through a semi-structured interview with women about their experience of care. An extraction form was used by a separate team of data collectors to record the background information of the enrolled women through the patient case file and delivery registers. The two teams were independent of each other and sent their collected data to the data coordinator separately.

*Data management-* The data were collected using a paper-based form. Data coordinators at each site verified the completeness of the forms and dispatched them in a sealed envelope each week via courier to the main research office. The forms were indexed and filed as per the sites, were reviewed for completeness at the central research office, and entered into a Census and Survey Processing (CS Pro) database. The data were then exported to a statistical package (SPSS software version 23) for further cleaning and analysis (S1 Data). The indicators from observation and interview forms regarding the care and services were mapped, and selected indicators were then considered for analysis. The data was extracted for only those cases which had complete data from all two information sources (observation and interviews). The data obtained from emergency cesarean sections as well as elective cesarean sections were excluded from this study.

## Respectful maternal and newborn care domains

Based on a review of the literature in respectful maternal and newborn care and available data from the NePeriQIP study [8,29,30], we considered three main domains of respectful care: 1) consent for care 2) respect and dignity of care, and 3) care provision. These domains were based on the framing that consent for care is fundamental to health care and unconsented care is a marker of disrespectful care; experience of respect and dignity during childbirth is critical to a positive experience; and care provision is the minimum standard.

## Mapping of observation checklist and interview questionnaires

The observation checklist and interview questionnaires were mapped and plotted in each of these domains. For *consent and counselling domain*, we found one question from the observation checklist and one semi-structured interview question were matched. For the *respect and dignity domain*, we found two questions from the observation checklist and two questions from semi-interview questionnaires. For the *care provision* domain, we found four questions in the observation checklist and five in the semi-structured interview questionnaires were matched. (Table 2).

*Socio- Demographic co-variates:* The study collected women's sociodemographic information, including maternal age; ethnicity (Dalit, Janajati, Madhesi, Muslim, Others) [31]; parity (nullipara, primipara, multipara), and maternal education (uneducated, primary, secondary or more). Wealth Index was also calculated: Using the principal component analysis, a wealth index was created based on the household asset and was categorized (poorest, poorer, middle, richer, and richest) [32].

*Statistical analyses-* Descriptive statistics of demographic characteristics of the study population in total and by hospital were done using an independent t-test for continuous variables such as age and gestation age and a one-way ANOVA test for categorical variables.

To assess the agreement between measurements (degree of concordance), we used two statistical methods [33]. The Cohen kappa (k) was used to assess the inter-measurement agreement for indicators within each domain. Cohen kappa takes into account the agreement by chance calculated as k = (observed agreement, Po- expected agreement, Pe)/(1- expected agreement, Pe). k statistics is expressed as a value between – 1–1 and is interpreted as a level of agreement as follows: 0% = agreement by chance, 10%-20% = slight agreement, 21–40% = fair agreement, 41%-60% = moderated

**Table 2. Domains and mapping of observation checklist and interview questionnaires.**

| Respectful care domains | Observation | Exit interviews questionnaires |
|---|---|---|
| Consent and Counseling | i) Was consent taken by the health workers with the mother on any intervention provided to the baby? (Consent Taken) | i) Were you adequately informed by the care provider about examinations, actions, and decisions taken for your care throughout the hospital stay? (Informed about Routine Care) |
| Respect/ dignity | i) Was the woman treated with kindness and respect during delivery? (Treated with respect and Dignity) | i) Were you treated with respect and was your dignity preserved during your stay at the hospital? (Treated with respect and Dignity) |
| | ii) Did anyone accompany the mother in the delivery room? (Companion During Birth) | ii) Did you have a companion of your choice during labor and childbirth? (Companion During Birth) |
| Care provision | i) Baby kept in the skin-to-skin contact in the mother's chest (Infant-Women Skin to Skin Contact) | i) Have you kept your baby's skin-to-skin contact immediately after birth? (Infant-Women Skin to Skin Contact) |
| | ii) Breastfeeding initiated (Initiation of Breastfeeding) | ii) Were the babies breastfed before transfer to the postnatal ward? (Initiation of Breastfeeding) |
| | iii) The Newborn's body and head are covered during the stay in the delivery room (Infant's Kept Warm) | iii) Were the newborns' bodies and heads covered after birth? (Infant's Kept Warm) |
| | iv) Anything applied to the cord? (Umbilical Care) | iv) Was anything applied to the umbilical stumps? (Umbilical Care) |

agreement, 61%-80% = substantial agreement; 81%-99% = near perfect agreement and 100% = perfect agreement. A kappa value of less than 60% indicates a significant level of disagreement. To present the results, we used heatmaps to visually show the level of agreement, with a heat maps color spectrum, Green = 75–100%; Yellow = 36–74% and 1–35% = Red.

The second method applied was the Bland-Altman plot, which compared two measurement techniques (observation and interview) for each domain, considering each domain as a continuous scale [34]. This method presents a scatter plot of the difference between the two measurements (Y-axis) against the average of the two measurements (X-axis). It displays a graph of bias (mean difference between two measurements) with 95% limits of agreement. Limits of an agreement are calculated as = mean difference between two measurements ± 1.96 X standard deviation of the observed difference.

Ethical Consideration- For this study, ethical approval (no. 26/2017) was received from the national ethical review board, Nepal Health Research Council (NHRC). Written consent was taken from the pregnant women who agreed to participate before the extraction and clinical observations. Surveillance officers were trained in maintaining the confidentiality of the information related to mothers and their newborns.

## Results

During the study period, a total of 30601 women were eligible for clinical observation of whom 1780 did not provide consent leaving 28821 women who consented and were observed. Among these 5979 had incomplete information or could not be matched with interview data, resulting in 22,832 women with matched observation records. Simultaneously, 34581 women who delivered at the hospitals were eligible for interview with 782 declining to participate leading to 33799 women who consented and were interviewed. Of these 10,967 had incomplete information or could not be matched with observation data, ultimately yielding 22,832 women with matched interview records. Thus, a total of 22,832 cases had matching information from both observation and interviews and were selected for final analysis. (Fig 1, S1 and S2 Tables)

The overall mean age of the delivered women was 23.79 ± 4.08. Most of the delivered women were 20–35 years. The women from the Brahmin/Chhetri ethnic group (n = 9085; 39.8%) were the largest group among the population followed by Janajati (n = 6749, 29.6%). Women with no previous birth (n = 12118; 53.1%) were the largest group among the population and women with 2 or more previous births (n = 2914, 12.7%) being the lowest. The women with education level secondary and above (n = 13580, 59.5%) were highest among the population. The women from richest group were lowest (n = 4869, 21.3%). (Table 3)

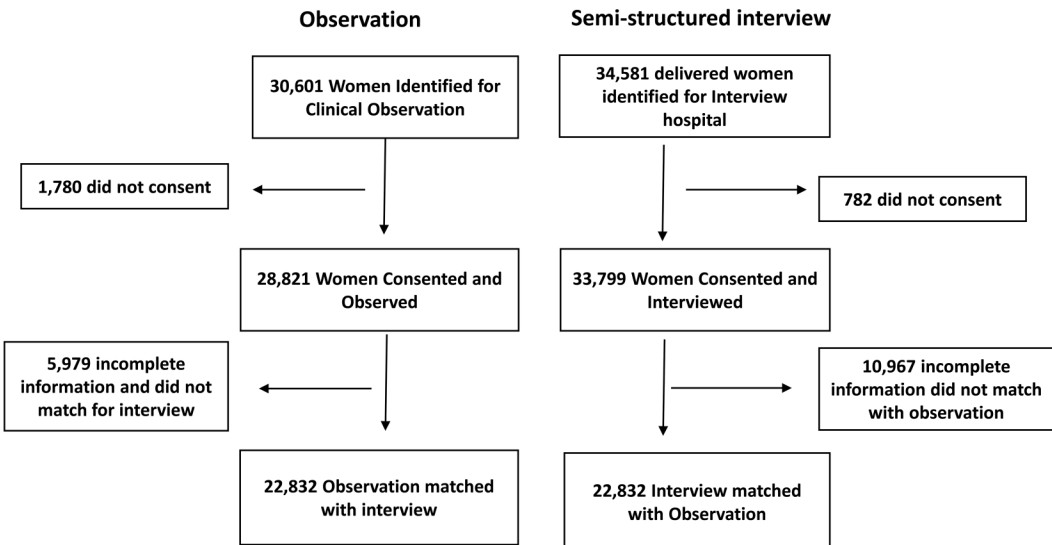

**Fig 1. Study Flow Chart.**

**Table 3. Background Characteristics of study population.**

| Indicators | Overall | Hospital A | Hospital B | Hospital C | p-value |
|---|---|---|---|---|---|
| **Age(Mean±SD)** | 23.79±4.08 | 23.89±4.16 | 22.99±3.81 | 24.2±4.0 | <0.0001* |
| < 20 years | 1634 (7.2) | 792 (8.3) | 425 (7.8) | 417 (5.3) | |
| 20- 35 years | 20994 (91.9) | 8610 (90.8) | 4963 (91.6) | 7421 (93.6) | |
| >35 years | 204 (0.9) | 84 (0.9) | 28 (0.5) | 92 (1.2) | |
| **Ethnicity** | | | | | <0.0001 |
| Dalit | 3034 (13.3) | 1271 (13.4) | 988 (18.2) | 775 (9.8) | |
| Janajati | 6749 (29.6) | 4108 (43.3) | 901 (16.6) | 1740 (21.9) | |
| Madhesi | 3212 (14.1) | 479 (5.0) | 2379 (43.9) | 354 (4.5) | |
| Muslim | 752 (3.3) | 151 (1.6) | 470 (8.7) | 131 (1.7) | |
| Chhetri/Brahmin | 9085 (39.8) | 3477 (36.7) | 678 (12.5) | 4930 (62.2) | |
| **Parity** | | | | | <0.0001 |
| 0 previous births | 12118 (53.1) | 4996 (52.7) | 2675 (49.4) | 4447 (56.1) | |
| 1 previous birth | 7800 (34.2) | 3317 (35.0) | 1810 (33.4) | 2673 (33.7) | |
| >2 births | 2,914 (12.7) | 1173 (12.3) | 931 (17.2) | 810 (10.3) | |
| **Education** | | | | | <0.0001 |
| No primary education | 2879(12.6) | 650(6.9) | 1758(32.4) | 471(6.0) | |
| Primary education | 4329 (19.0) | 1872 (19.7) | 1658 (30.6) | 799 (10.1) | |
| ≥Secondary education | 15,624 (68.5) | 6,964 (73.4) | 2000 (36.9) | 6660 (84.0) | |
| **Wealth index** | | | | | <0.0001 |
| Poorest | 7866(34.5) | 4036(42.5) | 2396(44.2) | 1434(18.1) | |
| Poorer | 4624(20.3) | 2238(23.6) | 1739(32.1) | 647(8.2) | |
| Middle | 2016(8.8) | 971(10.2) | 631(11.7) | 414(5.2) | |
| Richer | 3457(15.1) | 1196(12.6) | 479(8.8) | 1782(22.5) | |
| Richest | 4869(21.3) | 1045(11.0) | 171(3.2) | 3653(46.1) | |

One-Way ANOVA Test*, Chi-Square Test.

In the domain of consent and counseling, 47.3% of women were observed to have consented to care, while 78.8% of women reported being informed about routine care (k, LOA = 59.1%), There was a poor concordance between the two methods for consent for care. In the domain of respect and dignity, 99.0% of women were observed to have been treated with respect and dignity and 96.4% of them reported to have been treated with respect and dignity (k, LOA = 95.4%). This indicates that the concordance was near perfect agreement >90% between observation and women's experience in treatment with respect and dignity and companionship during labor and childbirth. In the same domain, 5.7% of women were observed to have a companion during birth in observation, while 7.6% of women reported having a companion of choice during labor and birth (k, LOA = 88.4%). There was a near-perfect level of agreement between the two measurements for companions during labor and childbirth. In the domain of care provision, 3.9% of the infants were observed to be kept in skin-to-skin contact while 37.9% of women reported to have kept their infant skin-to-skin contact (k, LOA = 61.7%). The concordance of provision of care for skin-to-skin contact had a moderate level of agreement between measurements. In the same domain, initiation of breastfeeding was observed in 11.2%, and it was reported by 11.3% (k, LOA = 84.9%). There was a high concordance with near-perfect agreement for breastfeeding between the two measurements. In the same domain, 87.1% of infants were observed to have umbilical care and 19.9% of women reported that the infant received umbilical care (k, LOA = 21.6%). There was poor concordance with a low level of agreement for umbilical care between the two measurements (Table 4).

In terms of disaggregation by hospitals, in Hospital A, in the domain of consent and counseling, 84.8% of women were observed to have consented to care, while 3.7% of women reported being informed about routine care (k, LOA = 17.5%). In the domain of respect and dignity, 2.3% of women had a companion during birth in observation, while 10.9% of women reported having a companion of choice during labor and birth (k, LOA = 87.3%). In the domain of care provision, 87.0% of infants were observed to have umbilical care and 15.9% of women reported that the infant received umbilical care (k, LOA = 36.6%). in Hospital B, In the domain of consent and counseling, 25.0% of women were observed to have consented to care, while 69.0% of women reported being informed about routine care (k, LOA = 39.1%). In Hospital C, in the domain of care provision, 6.2% of the infants were observed to be kept in skin-to-skin contact while 89.9% of women reported to have kept their infant in skin-to-skin contact (k, LOA = 13.3%). Even though the concordance between the two measurement approaches varied by hospitals, the level of agreement for each domain remained at a similar level. All hospitals had a high level of agreement for the respect and dignity domain, a poor level of agreement for the consent and counseling domain, and variation in the level of agreement for the provision of care (S3 Table).

Using the Bland Altman plot, the average difference between observed and reported measures for consent/counselling was -21.5% indicating that the reported measures were generally lower than the observed measures (Fig 2a). The average difference between observed and reported measures for treated with respect and dignity was -2.3% (Fig 2b). In the

**Table 4. Domains, Data collection method and item, and percentage agreement between observation and interview.**

| Respectful care domains | Observation (22,832) | Exit Interview (22,832) | Observed | Interviewed | K value, level of Agreement (%) |
|---|---|---|---|---|---|
| Consent/ counseling | Consent Taken | Informed about Routine Care | 47.30% | 78.80% | 59.10% |
| Respect/ dignity | Treated with respect and Dignity | Treated with respect and dignity | 96.40% | 99.00% | 95.40% |
| | Companion during birth | Companion of choice during labor-birth | 5.70% | 7.60% | 88.40% |
| Care provision | Infant-Women skin-to-skin contact | Infant-Women skin to skin Contact | 3.90% | 37.90% | 61.70% |
| | Initiation of Breastfeeding | Initiation of Breastfeeding | 11.20% | 11.30% | 84.90% |
| | Infant Kept warm | Infant's Kept Warm | 94.70% | 96.50% | 93.50% |
| | Umbilical care | Umbilical care | 87.10% | 19.90% | 21.60% |

Heat maps color spectrum, Green= 75-100%; Yellow = 36-74%; 1-35% = Red.

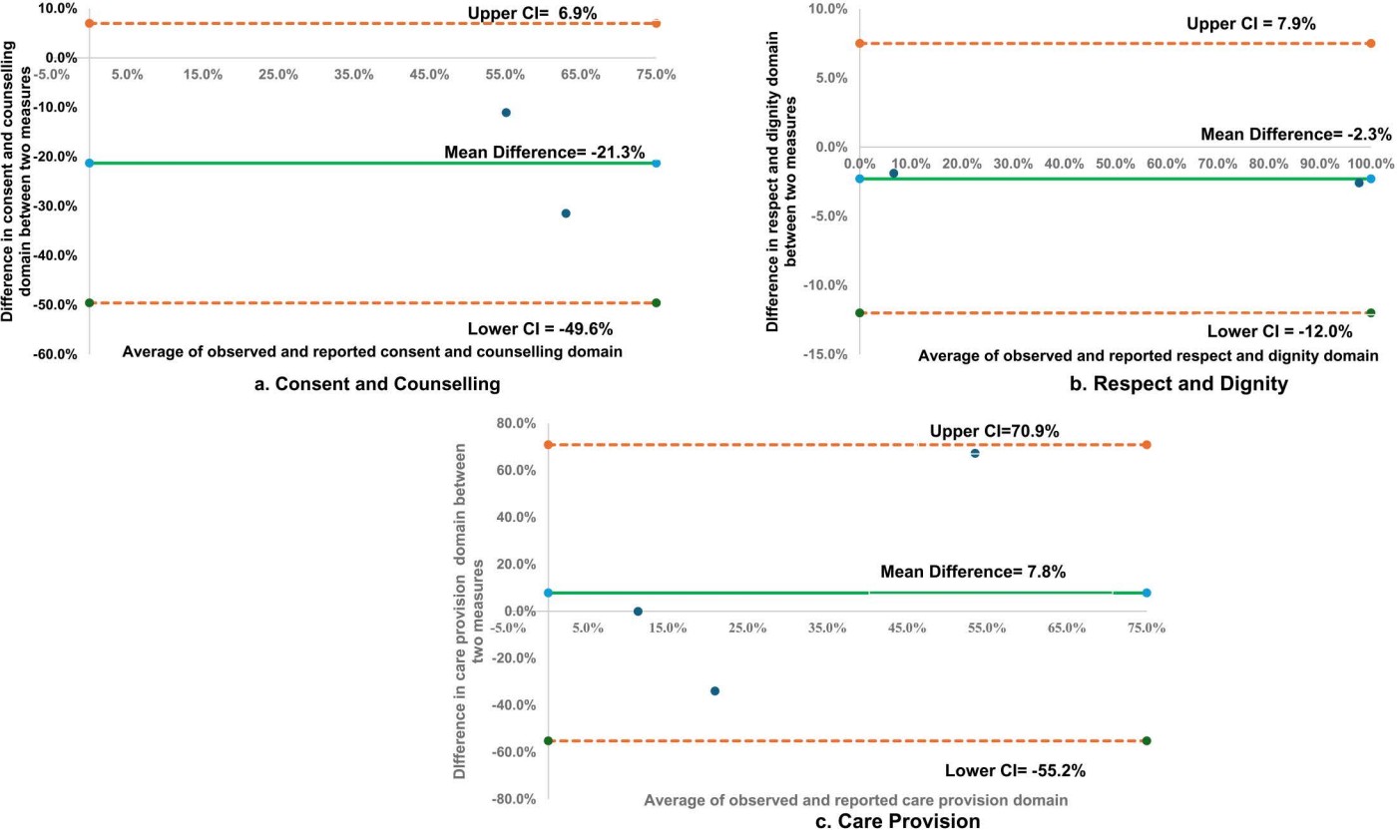

**Fig 2. Bland Altman Plots on average difference between observed and reported data in three domains.**

domain of care provision, average difference between observed and reported measures skin to skin contact was -7.8% (Fig 2c).

## Discussion

The concordance between women's reported experience and independent observation varies by different RMNC domains. There is discordance between the experience of care for consent and counseling (such as women consenting for newborn medical procedures) and provision of care (such as initiation of skin-to-skin contact immediately after birth), while the concordance is high for treatment with respect and dignity. Even though the correlation between the two measurement approaches varied by hospitals, the level of agreement for each domain remained at a similar level.

In terms of the poor level of agreement for consent and counseling, the observation showed less than half of women had proper counseling addressed. Some discordance is expected since the questions were not exactly the same in the observation checklist and interview questionnaire. An observer may have seen some counseling occur, but women may have felt it was inadequate or was rushed. A cross-sectional survey among women in 81 health facilities in Northern India, a setting similar to Nepal, showed a poor level of agreement between measures related to counseling, namely concerns being addressed by healthcare providers [35]. A multi-country study in Bangladesh, Ghana, and Tanzania which assessed the adoption of respectful maternity care in health facilities showed differences in consent and counseling during childbirth in the baseline period [36]. In the multi-country study, 61.1% of the women were observed to have been well informed on the care while 50.3% of women reported having been well-informed.

Our study showed a high level of agreement for respect and dignity demonstrated by healthcare providers in both observations and interventions, which is different than findings in other settings. An implementation research study conducted in two health facilities in Tanzania [37] found that the observers reported more disrespect and abuse than the women themselves. At baseline, across facilities, the prevalence of any observed disrespect and abuse were 69.8% and on self-report, 9.9%. A cross-sectional study conducted in nine public health facilities in Ibadan, Nigeria compared observed vs reported respectful maternal care in a 15-item scale [26]. The study showed very weak agreement between the two measures, with a difference between the reported and observed scores at an average of 41.1 percentage points. This could be related to the challenge of measuring disrespect and abuse, given that different individuals may have different interpretations and definitions [38]. Many practices that have been normalized in health care settings may not be recognized by patients or families as disrespectful, although observers may be trained to record them as such.

The poor level of agreement for the provision of care domain in our study, especially with umbilical care, is noteworthy since women perceived poor care while the observation showed standard umbilical care provision. More investigation around why this might have happened will be needed including understanding the health literacy of the women toward standard umbilical care and where umbilical care is taking place. This could also be related to women's experiences and interpersonal interactions, where clinical actions are occurring, but women feel that the care is substandard because it is not being explained or done is a thorough or careful way. Overall, we found that provision of care immediately after birth, such as skin-to-skin contact, had a poor level of agreement between approaches. This discordance could be due to the health literacy on understanding skin-to-skin contact with infants, and the challenge of defining and documenting (for example, the amount of time an infant is held in a skin-to-skin position and if it is sufficient). The EN-BIRTH multi-country study in five public hospitals in Bangladesh, Nepal, and Tanzania found similar results for early initiation of breastfeeding, skin-to-skin within 1 h of birth, and delayed cord clamping [39]. Agreement of early initiation of breastfeeding with skin-to-skin contact was moderate/high (69.5–93.9%) at four sites, but fair/poor for delayed cord-clamping (47.3–73.5%) in the EN-BIRTH study.

## Research gaps and priorities

The concordance and discordance between the two measurement techniques in different domains of respectful care indicate a research gap in the measurement of respectful maternal and newborn care. For example, the adequacy of information provided to pregnant women and concerns addressed during consent may vary by individual (individual bias), therefore the reporting of the experience of care needs validation through the direct observation method. However, observation may only state if information during consent took place but not whether the woman felt it was adequate. Thus, there may be cases where observation can be a complementary method to interview or vice versa. However, there is a paucity of evidence on the comparison between provision and experience of care using observation and interview techniques. Even though both provision and experience of care are two pillars of quality of care [3], there is a lack of recommended techniques for measurement together. As shown in the study, observation and interview techniques complement each other's findings, but there is a need for better tools for collecting data around RMNC that consider both qualitative and quantitative methods [40]. Our findings support the need to have multiple approaches or mixed methods designs when evaluating RMNC whereby both direct observations are used in addition to self-reported experiences of care [41]. There is also a need to create and use standard definitions where possible and have clear criteria for monitoring care provision and progress toward respectful care goals.

## Strengths and weaknesses of the study

A large sample size with more than 22,000 matched observations and interviews for respectful maternal and newborn care is a strength of the study. Further, the design of this study using three domains related to RMNC is a unique aspect

of this study, compared to ones that focus only on consent or quality. However, the primary study was not designed to correlate the observation checklist to the interview questionnaire. The primary study aimed to assess the experience of care using two different measures but did not assess the reliability of the observation checklist with interview questionnaires in its design phase [42]. Therefore, the observation and interview question wording were not identical and not designed to be complementary. Future research that primarily examines data collection methods should consider aligning indicators in the different tools as closely as possible during the study design phase to enable a more direct comparison. The study had some potential biases, such as a large sample of women who were excluded since they did not participate in both data collection approaches. To mitigate the potential risk of interviewer bias, the interviews were conducted by independently trained research nurses who were Nepali and spoke the local language, however there could have still been some detection bias.

### Global public health implication

The measurement of respectful or experience of care is largely absent in routine health information systems and periodic large-scale surveys, such as Demographic Health Surveys (DHS) or Multiple Cluster Indicator Surveys (MICS) [43,44]. Existing assessments primarily focus on provision of care, while experience related aspects such as consent and counseling, respect and dignity, and receipt of adequate information has largely been through qualitative studies rather than standardized epidemiological measurement. Metrics for experience of care and respectful maternal and newborn care are not yet standardized and validated measures and thus measuring experience of care is still aspirational in routine system and surveys. However, experience of care is a critical aspect of quality of care as it determines not only trust towards health care institution but also influences the mental wellbeing of patients and families, and has links with clinical outcomes as well [45].

To ensure a more accurate and actionable measurement of experience of care, it is necessary to compare different approaches to data collection. This study demonstrates that women's and newborn's reported experiences vary depending on the measurement methods used, indicating that relying solely on one approach may provide an incomplete or biased picture of care quality.

With further research and pilot testing, the integration of multiple measurement approaches such as hospital exit interviews alongside periodic independent observations can enhance routine monitoring of experience of care. Routine health information systems are also limited in terms of measuring key interventions necessary in maternal and newborn care, e.g., immediate skin-to-skin, cord care, etc., but efforts should be undertaken to include measures related to quality. The DHS have added provision of care indicators in recent versions, such as cord care [46], our study underscores the need to measure these large-scale survey findings through cross-sectional observational studies to ensure accuracy and completeness in routine data collection.

### Conclusion

This study identified significant variations between observed and self-reported measures of maternal and newborn care specifically in the domains of consent and counseling, respect and dignity, and provision of care. While the domain of respect and dignity showed high concordance between observations and exit interviews, consent and counseling, exhibited poor agreement, indicates discrepancies in how these aspects of care are perceived and reported. Similarly, within the provision of care domain, variations were noted in key indicators such as skin-to-skin contact umbilical care highlights challenges in accurate measurement. There is need to integrate standardized metrics and independent observational assessments to strengthen the accuracy of experience-of-care measurements and improve the monitoring and delivery of maternal and newborn health services. Further, this study highlights the need for a measurement approach that incorporates independent observations alongside self-reported data to improve maternal and newborn care assessments.

## Supporting information

**S1 Data. Dataset.**
(CSV)

**S1 Table. Background characteristics comparison between selected and unselected samples (Observation).**
(DOCX)

**S2 Table. Background characteristics comparison between selected and unselected samples (Interview).**
(DOCX)

**S3 Table. Domains, Data collection method and item, and percentage agreement between observation and interview by hospital.**
(DOCX)

**S1 File. S1_Checklist.**
(DOCX)

## Acknowledgments

We want to acknowledge and thank all the data collectors for the data collection and management.

## Author contributions

**Conceptualization:** Emma Sacks, Mary V. Kinney, Ashish KC.

**Data curation:** Omkar Basnet.

**Formal analysis:** Omkar Basnet.

**Investigation:** Emma Sacks, Mary V. Kinney.

**Methodology:** Emma Sacks, Mary V. Kinney, Ashish KC.

**Resources:** Omkar Basnet.

**Software:** Omkar Basnet.

**Supervision:** Emma Sacks, Mary V. Kinney.

**Validation:** Omkar Basnet, Mary V. Kinney.

**Visualization:** Omkar Basnet.

**Writing – original draft:** Omkar Basnet, Ashish KC.

**Writing – review & editing:** Emma Sacks, Mary V. Kinney, Ashish KC.

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
