## [Decision Letter · Decision Letter 0]

PGPH-D-24-01548

Respectful maternal and newborn care in Nepal: comparing linked observation and interview data to understand associated practices and experiences

Dear Dr. KC,

Thank you for submitting your manuscript to PLOS Global Public Health. After careful consideration, we feel that it has merit but does not fully meet PLOS Global Public Health’s publication criteria as it currently stands. Therefore, we invite you to submit a revised version of the manuscript that addresses the points raised during the review process, particularly the conceptual and methodological concerns around the purpose of the analysis, the capacity of the selected indicators to provide a robust basis for comparison, and the clarity of the results as currently presented.

We look forward to receiving your revised manuscript.

Kind regards,

Hannah Hogan Leslie, PhD

Academic Editor

Journal Requirements:

2. We have amended your Competing Interest statement to comply with journal style. We kindly ask that you double check the statement and let us know if anything is incorrect. 

3. Please provide separate figure files in .tif or .eps format.

Additional Editor Comments (if provided):

Please note the important reviewer feedback below. I would note a number of related concerns requiring consideration if you choose to revise and resubmit.

1) 'Respectful care' domains: the domains presented are not well situated within the WHO standards and other major guidelines and frameworks for quality of care, and it is not clear how these domains align with the articles provided as references. A more clear typology and more detail on the links between the domains measured here and the substantial literature on maternal care quality is required. In particular, care provision should be situated under Experience of Care (per WHO framework) or technical quality of care. It is particularly important to distinguish between experience of care measures and technical care delivery measures when approaching a measurement analysis. Whether the other two domains can both be described as respectful care vs the broader 'experience of care' grouping used in WHO standards should also be considered and appropriately situated within the literature.

2) Purpose of the analysis: This article rests on a body of literature comparing observed vs. self-reported technical care indicators (typically attempting to validate the reported indicator against the direct observation: https://pubmed.ncbi.nlm.nih.gov/33765956/) and a more recent literature on experience of care measurement. Assessments of experience measures have differed in whether to consider the patient perspective (https://pubmed.ncbi.nlm.nih.gov/31384074/,
https://pubmed.ncbi.nlm.nih.gov/37479486/) or the direct observation (https://www.joghr.org/article/38745-measurement-of-respectful-maternity-care-in-exit-interviews-following-facility-childbirth-a-criterion-validity-assessment-in-nigeria). (Disclosure - i am a co-author on Mehrtash et al.). This makes it particularly essential to address the reason for this research study and how the analysis matches that. The authors primarily refer to agreement and concordance, suggesting neither measurement approach is favored as more accurate, although the introduction refers to the need to validate self report. Any of these approaches could be appropriate, but a clear choice, justification, and alignment of methods accordingly is required.

3) Alignment of indicators: as noted in both reviews, a reliability analysis demands equivalent measurement between the two approaches; the items here do not all meet this standard. Please consider focusing the analysis only on indicators where a reasonable argument can be made for a common underlying action being measured.

4) Inclusion criteria: please include a comparison of characteristics of women whose data could be included in this analysis to those excluded (no match between observation and interview) at minimum as a supplement and consider any implications for your findings.

Reviewers' comments:

Reviewer's Responses to Questions

**Comments to the Author**

1. Does this manuscript meet PLOS Global Public Health’s publication criteria?

Reviewer #1: Partly

Reviewer #2: Partly

2. Has the statistical analysis been performed appropriately and rigorously?

Reviewer #1: Yes

Reviewer #2: Yes

3. Have the authors made all data underlying the findings in their manuscript fully available (please refer to the Data Availability Statement at the start of the manuscript PDF file)?

Reviewer #1: Yes

Reviewer #2: Yes

4. Is the manuscript presented in an intelligible fashion and written in standard English?

Reviewer #1: Yes

Reviewer #2: Yes

Reviewer #1: Dear Dr. Leslie,

Thank you for sharing this very interesting paper in which the authors take on the difficult challenge of comparing observed and patient-reported user experience to assess concordance of the two measurement approaches. The paper takes advantage of a large dataset with two overlapping modes of data collection and uses an approach that is highly important for improving quality measurement, but rarely available. However, the paper could be strengthened by more clearly laying out the motivation for comparing these two approaches and by providing a fuller discussion of the factors that might contribute to observed discordance. I share some major and minor comments for consideration below.

Major comments

The introduction should clearly articulate why the authors undertook the analysis. The stated aim is to evaluate the correlation between approaches, but why? What is to be learned or gained? My assumption would be the goal is to assess the reliability of patient report, but the many differences in measurement make this very difficult. If anything, this study demonstrates the challenges inherent in trying to capture and compare observed actions and respondent perceptions and could be useful in parsing out the contributing factors.

If the goal is to assess whether patients can perceive good or poor experience when it is observed to have occurred or not occurred, I have concerns about the extent to which the observation indicators and the exit interview questions measure the same constructs. The observation versus exit interview questions in the consent and counseling domain seem particularly different. For example, I am unclear how "was consent taken" as an observation question measures the same construct as "were you adequately informed about exams...". A woman may have been consented, but also may not have felt it was adequate information or that it was done for all interventions. If items do not measure the same construct, we would expect some degree of discordance simply from the instrument.

In the respect/dignity domain, how do observers assess whether the woman was "treated with kindness and respect during delivery"? That assessment could fall prey to the same murky measurement issues as exit interviews. Similarly, the observers assessed if anyone accompanied the mother, while the exit interviewer asked about companion of your choice. Again, quite a difference (and peculiar that more women said they had a companion than was observed.).

In the care provision domain (which to me reads more like technical quality than user experience), the observation indicator does not mention immediate skin contact like the interview question, and immediate can mean different things to different respondents. Helpfully, the care provision domain questions have the most fidelity between observations and interviews in the questions asked but have the least concordance, so this does suggest the analysis may be picking up some real differences.

Generally, more information is needed around how these items were observed and how questions were asked, including what explanatory information might have been available to respondents, especially for concepts like respect. This is essential for the viability of comparing these measures. While the authors mention some of these considerations in the discussion, they don't fully grapple with the extent to which small differences in measurement might influence the observed discordance. The discussion could be strengthened by elucidating the mechanisms by which disagreement could arise, e.g., differences in what was observable/perceptible, differences in how questions were asked, differences in expectations, issues with recall and reporting, etc. What are the possible factors contributing to discordance even if these can't be teased out in this data?

Minor comments

The title phrase "associated practices and experiences" is not clear to me--associated with what? My suggestion would be something around concordance of observed versus patient-reported measures of user experience, but I leave this to the authors' and editor's discretion.

It's unclear which of these clinical items (in the care provision section in particular) is observable by women in the study. Umbilical care had the lowest agreement--for example, is it possible that the observation team was able to see this done while mothers were not?

How were primary study participants selected for observations versus interviewing versus both? Were all respondents interviewed? Fig 1 suggests no. I am wondering whether the sample of women with both an observation and an interview could be different from the sample of women with only an observation, and how. This is mentioned in the limitations but details are not given.

What was the context in which women were interviewed? Given how many interview-reported items are higher than observed items, I wonder about the extent of courtesy bias, a common issue with exit interviewing.

The fig 2 plots are a bit hard to read. It looks like there may be an extra datapoint in fig 2a, and it's unclear in 2a what the "average 57.9%" text refers to. Fig 2b has different labels and fig 2c has none.

Reviewer #2: Background:

1. Add justifications for why "observations" and patient "experiences" need to correlate.

Methodology:

1. Describe the components of consent that were measured during "observation", if available. Did it include any of the components measured during "experiences"?

2. Comparisons of "experiences" and "observations" for consent and counseling does not seem appropriate. While the "observation" relates to consent, the "interview" is largely about counselling.

3. Describe how you calculated k. Did you use the "observed" proportion as the expected?

Results:

1. Figure 1 could be simplified by starting with 34,581 "interviewed" participants, followed by a subset of 30,601 that were "observed".

2. Table 4. Following from the methodology above, it is not clear why you are using 47.3% for each of the "interview" components.

3. Figure 2 (Bland Altman Plot). Better description of the figure is needed. There are 4 blue dots in 2a but only 3 measures are shown in Table 4.

Discussion

1. Describe the implications of using secondary data for analysis that are outside of its intended purpose.

2. Could there be other reasons for poor concordance between "observed" and "interviewed" other than differences in wording?

**Do you want your identity to be public for this peer review?** For information about this choice, including consent withdrawal, please see our Privacy Policy

Reviewer #1: No

Reviewer #2: No

---

## [Decision Letter · Decision Letter 1]

PGPH-D-24-01548R1

Respectful maternal and newborn care in Nepal: comparing linked observation and interview data

Dear Ashish,

Thank you for submitting your manuscript to PLOS Global Public Health. After careful consideration, we feel that it has merit but does not fully meet PLOS Global Public Health’s publication criteria as it currently stands. Therefore, we invite you to submit a revised version of the manuscript that addresses the points raised during the review process.

We look forward to receiving your revised manuscript.

Kind regards,

Collins Otieno Asweto, PhD

Academic Editor

Journal Requirements:

Additional Editor Comments (if provided):

Reviewer's Responses to Questions

**Comments to the Author**

Reviewer #1: (No Response)

Reviewer #3: All comments have been addressed

Reviewer #4: (No Response)

publication criteria?

Reviewer #1: Partly

Reviewer #3: Partly

Reviewer #4: (No Response)

3. Has the statistical analysis been performed appropriately and rigorously?

Reviewer #1: Yes

Reviewer #3: Yes

Reviewer #4: (No Response)

4. Have the authors made all data underlying the findings in their manuscript fully available (please refer to the Data Availability Statement at the start of the manuscript PDF file)?

Reviewer #1: Yes

Reviewer #3: Yes

Reviewer #4: (No Response)

5. Is the manuscript presented in an intelligible fashion and written in standard English?

Reviewer #1: Yes

Reviewer #3: Yes

Reviewer #4: (No Response)

Reviewer #1: Thank you to the authors for responses to the initial comments. A few remaining items require attention.

Centrally, the background section still does not articulate why comparing these methods is a worthwhile exercise, though I think that it is. What do we learn from his comparison and why conduct it? The edit from “aim” to “opportunity” dilutes the purpose further in my opinion. This should be a simple fix but I think it’s critical. The response to comment 2 from the editor is insufficient.

I have the same comment about the section on “global public health implications.” The section speaks to the importance of measuring experience for maternal and newborn health generally, but says little about the value of comparing different measurement approaches until the last sentence. This should be the central thrust of this section.

Certain domains that have been cut (eg, “concerns discussed” have been removed from table 2 and 4 but not deleted from the text in the discussion section (eg, paragraph 2). Please update.

The abstract specifically mentions how experience of care is difficult to validate. However, in their response, the authors make explicit this is not a validation study. I suggest removing this language if the study does not intend to validate a measurement approach.

Reviewer #3: The title does not fully provide a clear picture of the study. What is the importance of comparing the observation and interview data? The title is hanging and is not complete. Is it to assess or measure the quality of care with regards to "respectful maternal and newborn care". It can be rephrased.

Was the Nepal perinatal quality improvement program (NePeriQIP) a study or a national program? It is referenced as a program on page 4line 6-7. It is also referenced as a study on the same page under Methodology, line 16, and a project under line 18. There is need to clearly separate the NePeriQIP from the secondary analysis (basis of this manuscript) in the methodology.

The results section narrative lines 9-14 are not clearly written. It would be best to document the results for the observation checklist alone, then have another paragraph to explain the interview questionnaires (as shown in the figure 1). The figure explains the information better than the narrative.

On the last page of the comments (comment 9). The response of "might be due to interviewer bias" needs to be qualified further.

Reviewer #4: NA

**Do you want your identity to be public for this peer review?** For information about this choice, including consent withdrawal, please see our Privacy Policy

Reviewer #1: No

Reviewer #3: **Yes: ** Dr. Seraphine Kaminsa

Reviewer #4: No

---

## [Decision Letter · Decision Letter 2]

PGPH-D-24-01548R2

Measuring respectful maternal and newborn care in Nepal: comparing linked observation and interview data- prospective cohort study

Dear Dr. KC,

Thank you for submitting your manuscript to PLOS Global Public Health. After careful consideration, we feel that it has merit but does not fully meet PLOS Global Public Health’s publication criteria as it currently stands. Therefore, we invite you to submit a revised version of the manuscript that addresses the points raised during the review process.

We look forward to receiving your revised manuscript.

Kind regards,

Miquel Vall-llosera Camps

Staff Editor

Journal Requirements:

Reviewers' comments:

Reviewer's Responses to Questions

**Comments to the Author**

Reviewer #1: All comments have been addressed

Reviewer #4: All comments have been addressed

publication criteria?

Reviewer #1: Yes

Reviewer #4: Yes

3. Has the statistical analysis been performed appropriately and rigorously?

Reviewer #1: Yes

Reviewer #4: Yes

4. Have the authors made all data underlying the findings in their manuscript fully available (please refer to the Data Availability Statement at the start of the manuscript PDF file)?

Reviewer #1: Yes

Reviewer #4: Yes

5. Is the manuscript presented in an intelligible fashion and written in standard English?

Reviewer #1: Yes

Reviewer #4: Yes

Reviewer #1: No further comments.

Reviewer #4: In the abstract section you should describe those factors in relation to respectful maternity and new care

**Do you want your identity to be public for this peer review?** For information about this choice, including consent withdrawal, please see our Privacy Policy

Reviewer #1: No

Reviewer #4: **Yes: ** Muluken Tessema Aemiro

---

## [Decision Letter · Decision Letter 3]

Measuring respectful maternal and newborn care in Nepal: comparing linked observation and interview data- prospective cohort study

PGPH-D-24-01548R3

Dear Dr. KC,

We are pleased to inform you that your manuscript 'Measuring respectful maternal and newborn care in Nepal: comparing linked observation and interview data- prospective cohort study' has been provisionally accepted for publication in PLOS Global Public Health.

Best regards,

Julia Robinson

Executive Editor

Reviewer Comments (if any, and for reference):

Reviewer's Responses to Questions

**Comments to the Author**

Reviewer #1: All comments have been addressed

publication criteria?

Reviewer #1: Yes

3. Has the statistical analysis been performed appropriately and rigorously?

Reviewer #1: Yes

4. Have the authors made all data underlying the findings in their manuscript fully available (please refer to the Data Availability Statement at the start of the manuscript PDF file)?

Reviewer #1: Yes

5. Is the manuscript presented in an intelligible fashion and written in standard English?

Reviewer #1: Yes

Reviewer #1: (No Response)

**Do you want your identity to be public for this peer review?** For information about this choice, including consent withdrawal, please see our Privacy Policy

Reviewer #1: No
